# Distinct Expression of Inflammatory Features in T Helper 17 Cells from Multiple Sclerosis Patients

**DOI:** 10.3390/cells8060533

**Published:** 2019-06-04

**Authors:** Alessia Capone, Manuela Bianco, Gabriella Ruocco, Marco De Bardi, Luca Battistini, Serena Ruggieri, Claudio Gasperini, Diego Centonze, Claudio Sette, Elisabetta Volpe

**Affiliations:** 1Neuroimmunology Unit, IRCSS Fondazione Santa Lucia, 00143 Rome, Italy; a.capone@hsantalucia.it (A.C.); manu.b83@gmail.com (M.B.); gabrie84@gmail.com (G.R.); m.debardi@hsantalucia.it (M.D.B.); l.battistini@hsantalucia.it (L.B.); 2Department of Biology and Biotechnology Charles Darwin, Sapienza University, 00185 Rome, Italy; 3Department of Neuroscience “Lancisi”, San Camillo Hospital, 00152 Rome, Italy; serena.ruggieri@gmail.com (S.R.); c.gasperini@libero.it (C.G.); 4Unit of Neurology, IRCCS Neuromed, 86077 Pozzilli (IS), Italy; centonze@uniroma2.it; 5Institute of Human Anatomy and Cell Biology, Università Cattolica del Sacro Cuore, 00168 Rome, Italy

**Keywords:** multiple sclerosis, T helper -17 cells, interleukin-1 receptor, interleukin-2, interleukin-21, tumour necrosis factor-beta

## Abstract

Multiple sclerosis (MS) is a chronic inflammatory disease of the central nervous system (CNS). T helper (Th) 17 lymphocytes play a role in the pathogenesis of MS. Indeed, Th17 cells are abundant in the cerebrospinal fluid and peripheral blood of MS patients and promote pathogenesis in the mouse model of MS. To gain insight into the function of Th17 cells in MS, we tested whether Th17 cells polarized from naïve CD4 T cells of healthy donors and MS patients display different features. To this end, we analysed several parameters that typify the Th17 profile during the differentiation process of naïve CD4 T cells obtained from relapsing-remitting (RR)-MS patients (n = 31) and healthy donors (HD) (n = 28). Analysis of an array of cytokines produced by Th17 cells revealed that expression of interleukin (IL)-21, tumour necrosis factor (TNF)-β, IL-2 and IL-1R1 is significantly increased in Th17 cells derived from MS patients compared to healthy donor-derived cells. Interestingly, IL-1R1 expression is also increased in Th17 cells circulating in the blood of MS patients compared to healthy donors. Since IL-2, IL-21, TNF-β, and IL-1R1 play a crucial role in the activation of immune cells, our data indicate that high expression of these molecules in Th17 cells from MS patients could be related to their high inflammatory status.

## 1. Introduction

Multiple sclerosis (MS) is a disease affecting the central nervous system (CNS), and involves inflammation and neurodegeneration [1,2]. The symptoms of MS are due to the migration of inflammatory immune cells into the CNS and the destruction of myelin, which causes neuroinflammation and neurodegeneration [3]. In MS patients, pathogenic lymphocytes express molecules that facilitate their transit through the blood-brain barrier (BBB), which is normally precluded [2]. Once lymphocytes have penetrated the CNS, they proliferate and release cytokines, thus contributing to a complex inflammatory response that leads to myelin damage [4].

In particular, T helper (Th) 17 cells, a subtype of lymphocytes differentiated from naive CD4^+^ T cells and producing interleukin (IL)-17, are considered potent inflammatory effectors in MS [5,6]. Importantly, the pathogenic potential of Th17 cells does not rely exclusively on the production of IL-17, and it is believed that a combination of inflammatory factors is required [6]. For instance, Th17 cells express high levels of the C-C chemokine receptor 6 (CCR6) on the cell surface [7] that binds the C-C chemokine ligand 20 (CCL20) expressed by both Th17 cells and the vascular endothelium of the BBB, thus enabling the entry of Th17 cells into the encephalic compartment through the choroid plexus [8]. Moreover, Th17 cells produce other inflammatory cytokines, such as tumour necrosis factor (TNF)-α, IL-21, IL-22, which are likely to contribute to their pathogenicity [9,10].

Differentiation of Th17 cells requires IL-1β, IL-6, IL-23, and transforming growth factor (TGF)-β [11], which promote the expression of the RAR-related orphan receptor (ROR)-γt transcription factor and determine the activation of the Th17 cell lineage-specific differentiation program [12]. Among the Th17 polarizing cytokines, IL-1β is considered crucial for the inflammatory properties of human Th17 cells, due to its inhibitory role in IL-10 production [11,13]. In mice, IL-1β, together with IL-23, is implicated in the enhancement of Th17 cell differentiation primed by TGF-β and IL-6 [14], and for acquisition of a pathogenic phenotype by Th17 cells [9,15].

IL-1β binds a complex formed by its receptor (IL-1R1) and the IL-1R accessory protein (IL-1RAcP), thus triggering a signal transduction cascade involving the adaptor protein MYD88 and induction of specific inflammatory genes [16,17]. This pathway plays a key role in experimental autoimmune encephalomyelitis (EAE), the mouse model of MS, as a knockout of the IL-1R1 or the MYD88 gene in mice, which significantly ameliorates disease-associated phenotypes [18]. Importantly, their function in EAE seem to be related to Th17 cells, as ablation of IL-1R1 signalling strongly suppressed Th17-mediated EAE in cell transplantation experiments [15]. Moreover, it has been recently demonstrated that IL-1R1 expression is higher in CD4 T cells derived from MS patients in comparison to those from healthy donors [19]. 

In this retrospective study, we analysed expression of IL-1R1 and other Th17 typical features in Th17 cells derived from MS patients and healthy donors. In fact, a systematic study investigating the differentiation process of human Th17 cells in MS patients has never been performed. We hypothesised that Th17 cell differentiation is altered in MS patients, thus leading to the acquisition of pathogenic features that contribute to persistent inflammation in MS. 

## 2. Methods

### 2.1. MS Subjects

Patients diagnosed with relapsing–remitting (RR)-MS (n = 31) according to the revised McDonald diagnostic criteria [20] were enrolled in the study. The demographic and clinical characteristics of the RR-MS patients included in the study for blood sampling are described in Table 1. All patients included in the blood study did not take immunomodulant or immunosuppressive compounds for at least 2 months before recruitment. 

As controls, we used blood from age and gender matched individuals (n = 28) without inflammatory or degenerative diseases of the central or peripheral nervous system. These subjects were volunteers that underwent blood testing. 

Approval by the ethics committee of the IRCCS Neuromed, Pozzilli (IS), Italy and San Camillo Hospital, Rome (Italy), and written informed consent in accordance with the Declaration of Helsinki from all participants, were obtained before the study was initiated. 

### 2.2. Purification of Naive CD4^+^ T Lymphocytes from Adult blood

Peripheral blood mononuclear cells (PBMCs) were isolated by Ficoll gradient centrifugation (GE Healthcare, Little Chalfont, UK) from whole blood. For sorting of IL-1R1^+^ cells, PBMC were stained with anti-human CD4-FITC (Miltenyi, Bergisch Gladbach, Germany) (1:100) and anti-human IL-1R1-PE (RnD, Minneapolis, MN, USA) (1:20) and sorted by a MoFlo high speed cell sorter (Beckman Coulter, Atlanta, GA, USA). For sorting of naive CD4 T cells, the cells were stained with the anti-CD4-FITC (Miltenyi) (1:100), and CD4^+^ T Lymphocytes were purified by immunomagnetic selection, using the anti-FITC isolation kit (Miltenyi). After the isolation, the cells were stained with anti-CD4 FITC (Miltenyi) (1:100), anti-CD45RA BV421 (BD Biosciences, San Jose, CA, USA) (1:60), anti-CD45RO PE (BD Biosciences) (1:30), anti-CD27 APC (Miltenyi) (1:60), and CD4^+^ naive T cells were sorted by MoFlo high speed cell sorter (Beckman Coulter) as CD4^high^, CD45RA^high^, CD45RO^-^ and CD27^+^. Sorted cells had a purity of over 97%, measured by flow cytometry (data not shown).

### 2.3. Th Cell Differentiation Assay 

Naive CD4^+^ T cells were cultured in 96-well round-bottomed plates (Corning, New York City, NY, USA) at a density of 5 × 10^4^ per well in X-VIVO 15 serum-free medium (Lonza, Walkersville, MD, USA) in the presence of Dynabeads CD3-CD28 T cell expander (one bead per cell; Life Technologies, Carlsbad, CA, USA) and indicated cytokines: IL-1β (10 ng/mL), IL-6 (20 ng/mL), TGF-β (1 ng/mL) and IL-23 (100 ng/mL) (Miltenyi) for Th17 differentiation, as previously described [11,21]. After 5–6 days, cells were harvested and stained for flow cytometry analysis, or extensively washed, counted, and re-stimulated 1 × 10^6^ cells/mL with Dynabeads CD3-CD28 T cell expander (one bead per cell) for 24 h for cytokine quantification. 

### 2.4. Flow Cytometry Analysis

Cells were stained with the following antibodies: anti-human ROR-γt-BV421 (BD Biosciences) (1:20), anti-human CCR6-Alexa 647 (BD) (1:20), anti-human IL-1R1-PE (RnD) (1:20), anti-human CD4-PECy7 (Beckman Coulter) (1:100), CD161-BV421 (Biolegend, San Diego, CA, USA) (1:40), anti-human CD3 FITC (Miltenyi) (1:100). Samples were analysed using Cytoflex cytometer (Beckman Coulter, Brea, CA, USA) and analysed using FlowJo-10 software version 10.3.0. 

### 2.5. Cytokine Quantification

IL-17 in culture supernatant was quantified with an enzyme-linked immunosorbent assay (ELISA) kit (RnD Systems, Minneapolis, MS, USA). Other cytokines (IL-2, IL-4, IL-5, IL-7, IL-8, IL-9, IL-10, IL-13, IL-15, IL-21, IL-22, TNF-α, TNF-β, GM-CSF, IFN-γ, PDGF-AA, PDGF-AB/BB, CCL20) were quantified using a magnetic bead panel (Millipore, Burlington, MA, USA), following the manufacturer’s protocol, and analysed by Luminex. 

### 2.6. Statistical Analysis

For pair-wise comparisons of different conditions from the same donors or different donors, we used a parametric two-tailed paired or unpaired t test, respectively. One-way ANOVAs was performed to analyse the main effects of two conditions on the dependent variables and their interactions. Data were presented as mean ± standard error (s.e.m). The p values (p) of 0.05 or less, were considered statistically significant.

## 3. Results

### 3.1. IL-21 Production by Th17 Cells Is Increased in MS Patients Compared to Healthy Donors

Given the pathogenic role played by Th17 cells in MS, we hypothesized that Th17 cells differentiated from MS patients and healthy donors would display a differential cytokine profile. To test this hypothesis, naive CD4^+^ T cells from healthy donors and RR-MS patients were differentiated in Th17 cells under in vitro culture conditions. We analysed the cytokines produced by polarized Th cells after 6 days of culture. First, we measured the production of ten Th profile-associated cytokines for the Th1, Th17 and Th2 subsets. Interferon (IFN)-γ, the prototypical Th1 cytokine, and IL-22, whose expression pattern is closely related to that of IFN-γ [10], are included in the Th1 category (Figure 1A). The Th17 class of cytokines includes IL-17, the prototypical Th17 cytokine, IL-21, produced by Th17 cells and involved in the autocrine enhancement of their differentiation, and CCL20, the ligand of CCR6, which enables the entry of Th17 cells into the CNS (Figure 1B). A third set of cytokines produced mainly in Th2 conditions includes IL-4, IL-5, IL-13, IL-9, and IL-10 (Figure 1C). As expected, we observed the upregulation of Th17 cytokines and IFN-γ in Th17 compared to Th0 conditions, in both healthy donors and MS patients (Figure 1A,B). Typical Th2 cytokines (IL-4, IL-5, IL-13) were decreased in Th17 compared to Th0 conditions, while IL-9 and IL-10 were produced by Th17 cells, although not upregulated compared to Th0 (Figure 1C). While most of these cytokines were similarly modulated in MS and healthy donors (Figure 1A–C), we found that the expression of IL-21 was significantly higher in MS compared to healthy donors (Figure 1B), indicating a possible role for this cytokine in Th17-related MS pathogenesis.

### 3.2. Production of Cytokines Involved in Inflammation and T Cell Activation Are Increased in Th17 Cells from MS Patients

To investigate other features related to Th17 cells, we analysed the production of additional inflammatory cytokines that are typically produced by activated lymphocytes, including TNF-α, IL-8, and TNF-β, in Th17 cells differentiated from MS patients or healthy donors. We found that in MS patients, the production of TNF-α and β is significantly higher in Th0 and Th17 cells, respectively (Figure 2A). The production of IL-8, although more highly induced in Th17 compared to Th0 cells, was not differentially modulated in MS patients compared to healthy donors. 

Next, we analysed the production of three cytokines belonging to the common gamma-chain family, IL-2, IL-7, and IL-15, involved in regulating the expansion and activation of all T cell subsets. We observed that IL-2 is upregulated in both Th0 and Th17 cells from MS patients, whereas IL-7 is upregulated only in Th0 from MS patients and IL-15 is not modulated (Figure 2B). Given the role of IL-2 and IL-7 in T cell proliferation, these findings support the hypothesis of systemic T cell activation in MS patients.

To expand our analysis, we also tested expression of growth factors produced by T lymphocytes, such as platelet-derived growth factor (PDGF)-AA and AB/BB, and granulocyte-macrophage colony-stimulating factor (GMCSF). We observed that PDGF, either composed by subunit AA, AB or BB, is upregulated in Th17 cells, while an opposite trend was found for GM-CSF. However, both growth factors are not differentially modulated in MS compared to healthy donor Th17 cells (Figure 2C).

### 3.3. Th17 Cells Differentiated from MS Patients Express Higher IL-1R1 Than Those Differentiated from Healthy Donors

To address whether the acquisition of typical features of Th17 cells were differentially modulated in MS patients compared to healthy donors, we analysed the expression of the transcription factor ROR-γt, a master regulator of both mouse [12] and human Th17 cell differentiation [11,22], CCR6 [23] and IL-1R1 [15], which are not found in Th1 and Th2 cells, and are considered hallmarks of Th17 cells. This analysis revealed that ROR-γt and CCR6 are upregulated in Th17 cells from all individuals with no differences between cells obtained from MS patients and healthy donors (Figure 3A,B). In contrast, Th17 cells polarized from MS patients expressed significantly higher levels of IL-1R1 than corresponding Th17 cells polarized from healthy donors (Figure 3C). However, no significant differences were observed in cells obtained from MS patients in either the active or inactive phase of the RR disease (Appendix A), according to the presence (Gadolinium+) or the absence (Gadolinium-) of contrast-enhancing lesions detected by magnetic resonance imaging. This suggested that IL-1R1 expression on the Th17 cell surface was not influenced by the acute stage of inflammation. To determine whether IL-1R1 expression was dependent of a specific Th17-promoting cytokine, we removed them from the polarization medium. This analysis revealed that, similar to other Th17 molecules^11^, IL-1R1 expression in Th17 cells is mediated by the synergy between TGF-β and proinflammatory cytokines IL-1β, IL-6, and IL-23, although none of them exhibit a predominant role (Appendix A). 

### 3.4. IL-1R1^+^ Cells in the Blood Are Th17 Cells and Are Increased in MS Patients Compared to Healthy Donors

To investigate the physiological relevance of the increased IL-1R1 expression in Th17 cells differentiated from MS patients in vitro, we sorted blood memory CD4^+^ T cells for positivity or negativity in IL-1R1 expression. We found that CD4^+^ IL-1R1^+^ cells produced more IL-17 than CD4^+^ cells not expressing IL-1R1 (Figure 4A). This result indicates that IL-1R1 expression is also associated to IL-17-producing cells in vivo. In order to investigate whether Th17 cells are different in the blood of MS patients and healthy individuals, we analysed the frequency of circulating IL-1R1^+^ cells within the CD4^+^ CD161^+^ CCR6^+^ cell population, which corresponds to the population of memory Th17 cells [24,25]. Importantly, expression of IL-1R1 in the in vivo-differentiated Th17 cells was higher in MS patients compared to healthy donors (Figure 4B), indicating that mechanisms leading to overexpression of IL-1R1 in MS also occur in vivo. 

## 4. Discussion

In this study, systematic analysis of the typical features of Th17 cells revealed that cells derived from MS patients display higher expression of IL-1R1 compared to Th17 cells derived from healthy donors, and indicated that the corresponding higher expression of IL-1R1 previously observed in MS [19] patients is specifically associated to Th17 cells. Furthermore, we identified other inflammatory molecules that were differentially expressed that may contribute to the pathogenicity of Th17 cells in MS.

In fact, the analysis of a broad array of cytokines produced by Th17 cells in MS patients and healthy donors revealed that IL-21 is significantly upregulated in MS patients compared to healthy donors. This result is consistent with the high frequency of IL-21-producing T cells reported in both active and chronic parenchymal lesions of MS brain [26], and increased levels of IL-21 induced by anti-CD28 in CD4 from MS patients [27].

IL-21, similarly to IL-6 and IL-23, activates a signalling pathway involving STAT3, which directly binds the IL-17 and IL-21 promoters [28], thus contributing to the sustainment of Th17 lineage commitment [29]. However, IL-21 is also the most potent inducer of plasma-cell differentiation in vitro [30], and provides help to B cells [31]. Recently it has been reported that the B-T cell interaction plays a crucial pathogenic role in MS [32], which could be mediated by IL-21. Considering that MS involves infiltration of both T and B cells in parenchymal demyelinating lesions, and in lymphoid aggregates in the meninges [33,34], increased IL-21 levels produced by Th17 cells could play a critical role in MS. 

Our study also revealed that TNF-β, known as lymphotoxin-alpha (LTA), is upregulated in Th17 cells from MS patients, confirming previous observations showing that PBMC from RR-MS patients had increased expression of LTA [35]. Similar to LTA, which is involved in the regulation of cell survival and proliferation [36], other cytokines important for these functions in T cells, such as IL-2 and IL-7, are upregulated in MS, suggesting that they may synergize to promote T cell expansion during MS. In particular, IL-2 is significantly upregulated in cells derived from MS patients in both Th0 and Th17 conditions. These results are in agreement with the observed increase in IL-2 levels in the serum of patients with active MS [37], suggesting that the effector Th17 cells contribute to IL-2 production in MS. In fact, T regulatory cells, known to suppress effector immune functions, from subjects with RR-MS had impaired proliferation and reduced IL-2 secretion [38].

Another interesting finding of our study is that Th17 cells produce IL-8 and PDGF. Although they were not modulated in MS versus healthy donors, IL-8 and PDGF were not previously known to be produced by Th17 cells and are not typically considered Th cytokines. IL-8 plays an important role in inflammation due to its high neutrophil-attracting capacity [39], and could synergize with IL-17 in exerting this function. PDGF, which can be composed of two A subunits (PDGF-AA), two B subunits (PDGF-BB), or one of each (PDGF-AB) [40,41], plays a significant role in blood vessel formation. However, whether the production of PDGF contributes or not to the pathogenic functions of Th17 cells in MS will require direct investigation. 

Overall, the molecules upregulated in Th17 cells from MS patients could be related to their high sensitivity to IL-1β, one of the most important Th17-regulating cytokine. In fact, in the same set of experiments, IL-1R1 is significantly upregulated in MS patients, as previously demonstrated at transcriptional level in naïve and memory CD4 T cells, and in vitro differentiated Th17 cells [19]. 

We further analysed the expression of IL-1R1 on memory Th17 cells, which we found were significantly increased in MS patients. Previous comparative analysis on memory cells from MS and healthy donors was performed in Th1/Th17 cells, and revealed a reduced transcriptional expression of inflammatory factors, such as IFNG, CCL3, CLL4, and GZMB [42]. However, a broader analysis at protein level of Th17 and Th1/17 subsets isolated from MS patients may help to identify more disease-related features.

Our results indicated that IL-1R1 is upregulated at protein level in Th17 cells obtained from both in vitro and in vivo differentiated CD4 T cells of MS patients, and that IL-1R1 signalling is a critical step in the regulation of human Th17 cells in MS. In fact, it is already known that the IL-1β/axis is important for differentiation of naive CD4 T cells into Th17 cells [11,14]. Recently it has been demonstrated that IL-1β is also crucial for induction of IL-17A by memory CD4 T cells in the absence of T cell receptor (TCR) engagement, introducing a TCR-independent innate-like pathogenic role of Th17 cells mediated by IL-1β/IL1R1 signalling [43].

The relevance of IL-1R1 in MS is supported by previous results reporting that IL-1R1-deficient mouse T cells developed a delayed and less severe EAE, and a lower percentage of IL-17 producing cells in the inflamed CNS compared to wild type mice^15^. In conclusion, our study identifies IL-21, IL-2, TNF-β, and IL-1R1, as molecules likely involved in the mechanisms conferring pathogenicity to human Th17 cells and opens another perspective on the use of these molecules as therapeutic targets or biomarkers of MS disease.

## Figures and Tables

**Figure 1 cells-08-00533-f001:**
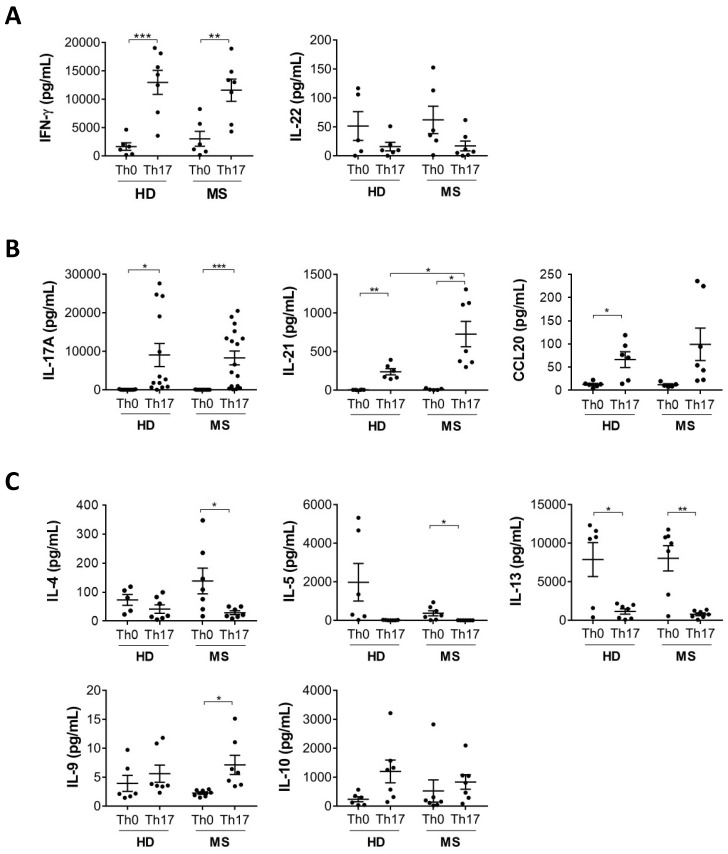
IL-21 production by Th17 cells is increased in MS patients compared to healthy donors. Naive CD4 T cells from healthy donors (HD) and multiple sclerosis (MS) patients were cultured with antiCD3-antiCD28 alone (Th0) or antiCD3-antiCD28 + TGF-β, IL-6, IL-23 and IL-1β (Th17). At 5 days of differentiation the levels of typical cytokines of Th1 (**A**), Th17 (**B**), and Th2 (**C**) cells were analysed by multiplex assay (Luminex) in cell supernatants (* *p* < 0.05; ** *p* < 0.01; *** *p* < 0.001).

**Figure 2 cells-08-00533-f002:**
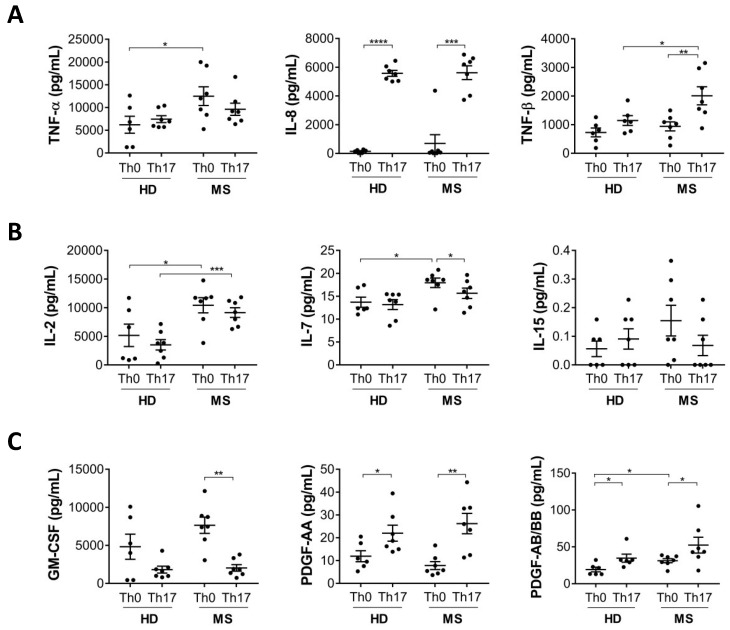
Production of cytokines involved in inflammation and T cell activation are increased in Th17 cells from MS patients. Naive CD4 T cells from healthy donors (HD) and multiple sclerosis (MS) patients were cultured with antiCD3-antiCD28 alone (Th0) or antiCD3-antiCD28 + TGF-β, IL-6, IL-23 and IL-1β (Th17). At 5 days of differentiation the levels of inflammatory cytokines (**A**), cytokines involved in T cell expansion (**B**), and growth factors (**C**) were analysed by multiplex assay (Luminex) in cell supernatants (* *p* < 0.05; ** *p* < 0.01; *** *p* < 0.001; **** *p* < 0.0001).

**Figure 3 cells-08-00533-f003:**
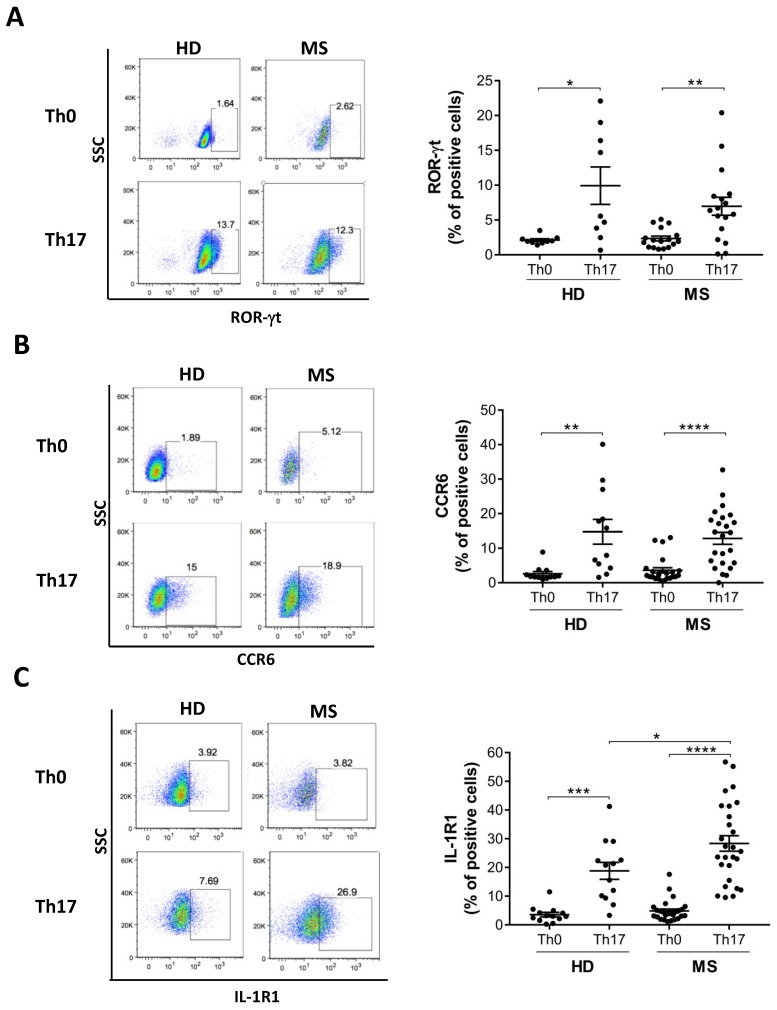
Th17 cells differentiated from MS patients express higher IL-1R1 than those differentiated from healthy donors. Naive CD4 T cells from healthy donors (HD) and multiple sclerosis (MS) patients were cultured with antiCD3-antiCD28 alone (Th0) or antiCD3-antiCD28 + TGF-β, IL-6, IL-23 and IL-1β (Th17). At 5 days of differentiation cells were stained with specific antibodies against RORγt (**A**), CCR6 (**B**), and IL-1R1 (**C**) and analysed by flow cytometry. Graphs represent the results of independent experiments. (* *p* < 0.05; ** *p* < 0.01; *** *p* < 0.001; **** *p* < 0.0001.).

**Figure 4 cells-08-00533-f004:**
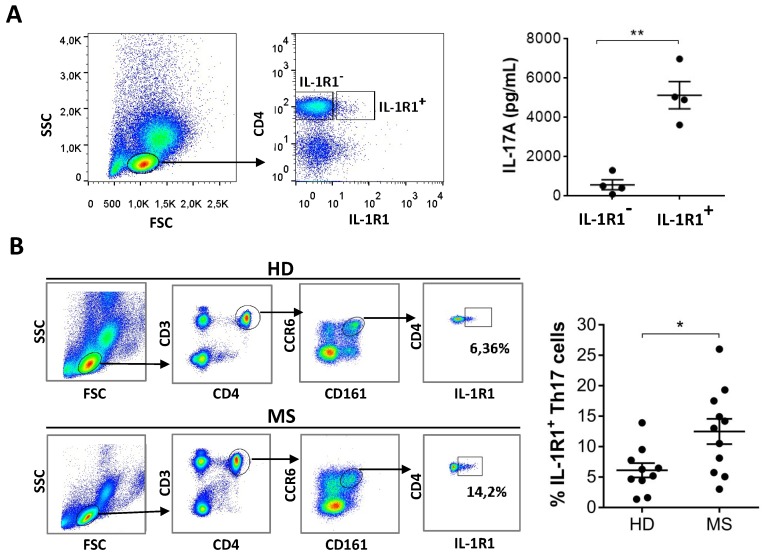
IL-1R1^+^ cells in the blood are Th17 cells and are increased in MS patients compared to healthy donors. Blood memory cells sorted as CD4^+^ IL-1R1^+^ and CD4^+^ IL-1R1^-^ cells (gating strategy in the left panel) were stimulated with antiCD3-antiCD28 for 24h. Supernatants were then analysed for IL-17 production by ELISA (**A**). Peripheral blood mononuclear cells from healthy donors (HD) and multiple sclerosis (MS) patients were stained with specific antibodies to analyse the frequency of CD3^+^ CD4^+^ CCR6^+^ CD161^+^ IL-1R1^+^ cells by flow cytometry (**B**). Graphs represent the results of independent experiments. (* *p* < 0.05; ** *p* < 0.01).

**Table 1 cells-08-00533-t001:** Demographic and clinical characteristics of MS subjects at the time of experiment.

**Number**	31
**Gender (male/female)**	5/26
**Age (years)**	43 ± 9.7
**EDSS**	2 ± 1.35
**MRI (gadolinium +/−)**	7/24

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
