# Peer review of "Distinct Expression of Inflammatory Features in T Helper 17 Cells from Multiple Sclerosis Patients"

_cells, 2019, doi:10.3390/cells8060533_

Round 1

Reviewer 1 Report

In this study, Capone et. al,  reaveled that IL-2, IL-21, TNF-β, and IL-1R are upregulated in in vitro differentiated Th17 cells obtained from MS patients compared to those obtained from healthy donors, indicating that these molecules could be related to their high inflammatory status in MS. Interestingly, they validated upregulation of IL-1R1 on freshly purified Th17 cells of MS patients compared to healthy donors, suggesting that common mechanisms are involved in in vitro and in vivo differentiation of Th17 cells. It is an interesting study and the results are clearly presented.

Minor comments:

In Figure 4b the gate of CCR6+ CD161+, corresponding to IL-17 producing cells, should be presented as a circle instead of a quadrant.

The size of the panels in figure1 must be increased.

The authors write IL-1R1 in the text and IL-1R in the figures, so they need to uniform the name of the receptors in text and figures.

According to the description of the results, in Figure 4a the labels IL-1b+/- must be replaced by IL-1R1 +/- .

Author Response

Response to Reviewer1

In this study, Capone et. al,  reaveled that IL-2, IL-21, TNF-β, and IL-1R are upregulated in in vitrodifferentiated Th17 cells obtained from MS patients compared to those obtained from healthy donors, indicating that these molecules could be related to their high inflammatory status in MS. Interestingly, they validated upregulation of IL-1R1 on freshly purified Th17 cells of MS patients compared to healthy donors, suggesting that common mechanisms are involved in in vitro and in vivo differentiation of Th17 cells. It is an interesting study and the results are clearly presented.

In Figure 4b the gate of CCR6+ CD161+, corresponding to IL-17 producing cells, should be presented as a circle instead of a quadrant.

We thank the reviewer for this suggestion. We modified the figure as suggested.

The size of the panels in figure1 must be increased.

We increased the size of the panels in Figure 1 and also in Figure 2.

The authors write IL-1R1 in the text and IL-1R in the figures, so they need to uniform the name of the receptors in text and figures.

We modified the name IL-1R in IL-1R1 in the revised version of figures.

According to the description of the results, in Figure 4a the labels IL-1b+/- must be replaced by IL-1R1 +/- .

We modified the labels as requested.

Reviewer 2 Report

Capone and colleagues investigated different cytokines of Th17 cells from MS patients vs. healthy donors. They found that Th17 cells from MS patients produce superior amounts of IL-21, TNF-Beta, IL-2, IL-1R1 compared to healthy controls.  

The findings presented in this article are interesting. I still think the manuscript needs major revisions in order to make it 1. More reader-friendly and 2. More transparent for the readers. Moreover, the article needs careful proof-reading again to increase overall accuracy of the phrasing. My concrete comments are below.

Abstract/Introduction:

Even though I generally agree, one must be careful in the wording of “MS is an autoimmune disease”. While several lines of evidence support that notion, one must still be accurate and acknowledge at least some uncertainty in these regards. The same is true for the next sentence: the primary target for immune cells in the CNS is not elucidated so far, and if myelin is attacked primarily is still under debate. “Myelin surrounding the axons” – there is no myelin not surrounding axons, thus, this seems redundant.

I like brief introductions. However, I recommend concisely adding some details on Th17 cells since not every reader might be familiar with that concept: which cells differentiate to Th17 cells? Th17 cells with regards to CD4/8 cells?

Myd88 vs. MYD88 – not consistent.

Neither from the abstract, nor from the introduction, I can get a grasp on the concrete motivation of this study. Quite some studies have addressed cytokine production from Th17 cells in MS patients. What would be the niche of your study? Please add a concrete aims and/or hypotheses to the introduction and abstract.

I would recommend removing the summary of your findings at the end of the introduction. This belongs to the first paragraph of the discussion.

State patient/control numbers also in the abstract.

Methods:

You should add to your methods that patients were diagnosed according to McDonalds criteria version 2010 rather than just referring to the reference.

Is this a retrospective study? Please state.

For which time interval did you recruit patients? Single-center?

I am not able to find table 1. Independently of that, add patient and control numbers to the method sections.

Add concentrations of used antibodies (1:x?).

What is a non-parametric t-test? T tests rely on Gauss distribution, are thus parametric. If you used a t test, how did you check for normal distribution?

Did you correct for multiple comparisons?

How were patients under relapsed defined?

Results:

Here, you nicely state your hypotheses – please do so in the introduction.

You have a weird blue bar in your Figure 1 panel C.

Please keep the referencing in your results section to a minimum – adding context to the results section can be helpful but, in your case, I feel it is not necessary to have all these references.

Discussion:

You mention PDGF with regards to blood vessel formation which contribute to disease progression in MS (around line 257): first, your reference (No 48) refers to CIS patients – not MS patients; second, the study from your reference measured cerebral blood flow but in MRI not vessel formation and you can have an increased blood flow by a simple vasodilation. Moreover, it is rather hypoxemia which is involved in MS lesions (virtual hypoxia – see work by Stys and colleagues). Thus, this referral seems a bit clumsy. Please also check your other references.

In your conclusion, you simply state that “molecules” are likely involved in MS – I recommend re-phrasing your conclusion to make it more unique to your paper and also to make it more stand-alone since some readers might only read certain parts of your manuscript.

Competing interests:

I would state “ …no additional financial conflict of interest”.

Author Response

Response to Reviewer 2

Capone and colleagues investigated different cytokines of Th17 cells from MS patients vs. healthy donors. They found that Th17 cells from MS patients produce superior amounts of IL-21, TNF-Beta, IL-2, IL-1R1 compared to healthy controls. 

The findings presented in this article are interesting. I still think the manuscript needs major revisions in order to make it 1. More reader-friendly and 2. More transparent for the readers. Moreover, the article needs careful proof-reading again to increase overall accuracy of the phrasing. My concrete comments are below.

Abstract/Introduction:

Even though I generally agree, one must be careful in the wording of “MS is an autoimmune disease”. While several lines of evidence support that notion, one must still be accurate and acknowledge at least some uncertainty in these regards. The same is true for the next sentence: the primary target for immune cells in the CNS is not elucidated so far, and if myelin is attacked primarily is still under debate. “Myelin surrounding the axons” – there is no myelin not surrounding axons, thus, this seems redundant.

We thank the reviewer and we modified the sentences deleting the words “autoimmune” and “myelin surrounding axons” (Line 37-39).

I like brief introduction. However, I recommend concisely adding some details on Th17 cells since not every readers might be familiar with that concept. Which cells differentiate to Th17 cells? Th17 cells with regard to CD4/8 cells?

We thank reviewer for the suggestion. We added details on Th17 cells at Line 44-45.

Myd88 vs. MYD88 – not consistent.

We modified Myd88 in MYD88 (Line 61).

Neither from the abstract, nor from the introduction, I can get a grasp on the concrete motivation of this study. Quite some studies have addressed cytokine production from Th17 cells in MS patients. What would be the niche of your study? Please add a concrete aims and/or hypotheses to the introduction and abstract.

We thank the reviewer for this important comment. We added our hypothesis in the abstract (Line 23-24) and Introduction (Line 71-73).

I would recommend removing the summary of your findings at the end of the introduction. This belongs to the first paragraph of the discussion.

We removed the summary of our findings from the end of introduction and we added to the first paragraph of the discussion (Line 227-232).

State patient/control numbers also in the abstract.

We added in the abstract the total number of patients and controls used for this study (Line 25-26).

Methods:

You should add to your methods that patients were diagnosed according to McDonalds criteria version 2010 rather than just referring to the reference.

We modified this sentence accordingly to reviewer’s suggestion.

Is this a retrospective study? Please state.

Yes, it is a retrospective study. We now state it at Line 69 of the revised manuscript.

For which time interval did you recruit patients? Single-center?

We recruited patients for two years from two hospitals: Department of Neuroscience “Lancisi”, San Camillo Hospital, Rome, Italy, and Unit of Neurology, IRCCS Neuromed, Pozzilli (IS), Italy. This is now stated at Line 98-99.

I am not able to find table 1. Independently of that, add patient and control numbers to the method sections.

We added Table 1 in the text (Line 124), and patient and control numbers to the Methods section (Line 76, 81).

Add concentrations of used antibodies (1:x?).

We added concentrations (1:x) of each antibody in the Methods section.

What is a non-parametric t-test? T tests rely on Gauss distribution, are thus parametric. If you used a t test, how did you check for normal distribution?

We apologize for the mistake. We used a parametric t-test assuming data as normally distributed. We removed the “non” (Line 120).

Did you correct for multiple comparisons?

We did not apply any correction.

How were patients under relapsed defined?

Patients were defined active or inactive according to the presence (Gadolinium+) or the absence (Gadolinium-) of contrast-enhancing lesions by magnetic resonance imaging. We added this sentence in the revised version of the manuscript (Line 71-73).

Results:

Here, you nicely state your hypotheses – please do so in the introduction.

We added a sentence in the introduction of the revised manuscript (Line 22-25) where we stated our hypothesis.

You have a weird blue bar in your Figure 1 panel C.

We removed the blue bar.

Please keep the referencing in your results section to a minimum – adding context to the results section can be helpful but, in your case, I feel it is not necessary to have all these references.

We thank the Reviewer for this suggestion. We have now kept only references that we believe to be essential to explain the rational of category assignment of cytokine IL-22 (Line 134)

Discussion:

You mention PDGF with regards to blood vessel formation which contribute to disease progression in MS (around line 257): first, your reference (No 48) refers to CIS patients – not MS patients; second, the study from your reference measured cerebral blood flow but in MRI not vessel formation and you can have an increased blood flow by a simple vasodilation. Moreover, it is rather hypoxemia which is involved in MS lesions (virtual hypoxia – see work by Stys and colleagues). Thus, this referral seems a bit clumsy. Please also check your other references.

As suggested by the Reviewer, we removed reference No48 and the related sentence.

In your conclusion, you simply state that “molecules” are likely involved in MS – I recommend re-phrasing your conclusion to make it more unique to your paper and also to make it more stand-alone since some readers might only read certain parts of your manuscript.

We rephrased the conclusion adding the name of the molecules found modulated in our study (Line 283-286).

Competing interests:

I would state “ …no additional financial conflict of interest”.

We modified the sentence as requested (Line 290).

Reviewer 3 Report

The manuscript „Distinct expression of inflammatory features in T helper 17 cells from multiple sclerosis patients“ (Capone R. et al.) represents a very interesting and well-designed study which characterize the process of Th17 cell polarization in multiple sclerosis (MS) patients in vitro and a secretory potential of their peripheral blood Th17 lymphocytes in vivo.

 For this purpose they quantified the production of cytokines in the supernatants of Th17 cells generated in vitro from naive CD4 T cells obtained from the blood of patients diagnosed as relapsing–remitting MS and those from healthy donors. The data obtained by flow cytometric analysis and ELISA revealed that after 6 days of cultivation Th17 cell from MS patients produced more IL-21, as well as more cytokines involved in inflammation and T cell activation, such as TNF-β, IL-2 and IL-1R1 than Th17 cells from healthy donors. Besides, they found that IL-1R1 expression was also increased in Th17 cells circulating in the blood of MS patients, compared to healthy donors

The data point to important signaling pathways through which Th17 cells may induce inflammation and demyelination in the brain. Owing to the novelty of the results and well documented state of the art in the discussion I would recommend the fast publication of this interesting study.

Minor objections are the following:

Line 139     compared to healthy donors (Figure 1C)…….-   should be (Figure 1B)

Fig. 1 C       IL-13 graft   …………          blue column on HD needs to be deleted

Line 164     composed by subunit AA, than AB or BB, is upregulated in Th17 cells.. (delete than)

Author Response

Response to Reviewer 3

The manuscript â€žDistinct expression of inflammatory features in T helper 17 cells from multiple sclerosis patients“ (Capone R. et al.) represents a very interesting and well-designed study which characterize the process of Th17 cell polarization in multiple sclerosis (MS) patients in vitro and a secretory potential of their peripheral blood Th17 lymphocytes in vivo.

 For this purpose they quantified the production of cytokines in the supernatants of Th17 cells generated in vitro from naive CD4 T cells obtained from the blood of patients diagnosed as relapsing–remitting MS and those from healthy donors. The data obtained by flow cytometric analysis and ELISA revealed that after 6 days of cultivation Th17 cell from MS patients produced more IL-21, as well as more cytokines involved in inflammation and T cell activation, such as TNF-β, IL-2 and IL-1R1 than Th17 cells from healthy donors. Besides, they found that IL-1R1 expression was also increased in Th17 cells circulating in the blood of MS patients, compared to healthy donors

The data point to important signaling pathways through which Th17 cells may induce inflammation and demyelination in the brain. Owing to the novelty of the results and well documented state of the art in the discussion I would recommend the fast publication of this interesting study.

Minor objections are the following:

Line 139     compared to healthy donors (Figure 1C)…….-   should be (Figure 1B)

Fig. 1 C       IL-13 graft   …………          blue column on HD needs to be deleted

Line 164     composed by subunit AA, than AB or BB, is upregulated in Th17 cells.. (delete than)

We thank the Reviewer for the comments and we have now corrected all the highlighted mistakes in the revised manuscript.

Round 2

Reviewer 2 Report

The authors have addressed my concerns regarding their manuscript. It adds now a further good piece of evidence on Th17 cell contribution in MS.

This manuscript is a resubmission of an earlier submission. The following is a list of the peer review reports and author responses from that submission.

Round 1

Reviewer 1 Report

In this study, authors reaveled that IL-2, IL-21, TNF-β, and IL-1R play a crucial role in the activation of immune cells, and their study indicated that high expression of these molecules in Th17 cells from MS patients could be related to their high inflammatory status. Also, authors reported that the IL-21 is significantly up-regulated in MS patients compared to control. In conclusion, the mentioned molecules are involved in the mechanisms conferring pathogenicity to human Th17 cells in MS patients. It is an interesting study and the results were clearly presented.

Minor comments:

Page 3; Line 75: Authors can give more details of the samples instead of referring other article for the details.

Authors showed the higher expression of IL-1R in MS patients. IL1RI is the receptor that mediates the biological actions of IL1, at the same time IL1RII is a decoy receptor that may serve to buffer the effects of excessive IL1 concentrations. Due to the functional difference of 1L1-RI and IL1-RII, authors could clarify the specific type of IL1-R overexpressed in the MS patients compared to normal patients.

Authors if could validate the important findings in the manuscript more than one techniques is preferable. Along with the protein expression, authors may show the corresponding gene expressions which will provide more validation. (Since authors have satisfactory levels of data, this recommendation won’t affect the acceptance of the manuscript).

Authors could improve the discussion section using most recent references (2017 and 2018).

Authors can also clarify if there are any dissimilarity results in the different age group of the patients? 

Page number 3: Authors could increase the size of the panels of figure 1, the letters looks small.

Author Response

Reviewer 1

Point 1: Page 3; Line 75: Authors can give more details of the samples instead of referring other article for the details.

Response to Point 1: At page 3 Line 75 we cited the article that define the criteria applied by clinical neurologists for the diagnosis of relapsing-remitting MS. We clarified that in the new sentence: “Patients diagnosed as relapsing–remitting (RR)-MS according to established criteria20 were enrolled in the study”. However, details of MS samples used in our study are described in Table 1. Moreover, since this is a very important issue, we moved the sentence: ”All patients included in the blood study did not take immunomodulant or immunosuppressive compounds at least 2 months before recruitment” at the beginning of the paragraph.

Point 2: Authors showed the higher expression of IL-1R in MS patients. IL1RI is the receptor that mediates the biological actions of IL1, at the same time IL1RII is a decoy receptor that may serve to buffer the effects of excessive IL1 concentrations. Due to the functional difference of 1L1-RI and IL1-RII, authors could clarify the specific type of IL1-R overexpressed in the MS patients compared to normal patients.

Response to Point 2: We thank the reviewer for this important suggestion. We used antibody against anti-IL1R1 and we modified the name IL-1R in IL-1R1 in the revised version of manuscript , figures, and supplementary data.

Point 3: Authors if could validate the important findings in the manuscript more than one techniques is preferable. Along with the protein expression, authors may show the corresponding gene expressions which will provide more validation. (Since authors have satisfactory levels of data, this recommendation won’t affect the acceptance of the manuscript).

Response to Point 3: We measured the levels of IL-1R1 at protein level, instead of transcriptional level. Since the amount of blood samples from MS patients does not permit to analyse the expression of IL-1R1 protein by both flow cytometry and western blot, we preferred flow cytometry analysis which allows to better quantify the levels of the protein and compare these levels among a high number of samples.   However, we validated our findings using two different approaches: Th17 cells generated in vitro from naive CD4 T cells and memory Th17 cells generated in vivo purified from the blood.

Point 4: Point Authors could improve the discussion section using most recent references (2017 and 2018).

Response to Point 4: We added recent references in the revised version of the manuscript (Line 275-280; Line 283-287; Line 238-239; Line 243-244):

Hu D, Notarbartolo S, Croonenborghs T, Patel B, Cialic R, Yang TH, et al. Transcriptional signature of human pro-inflammatory TH17 cells identifies reduced IL10 gene expression in multiple sclerosis. Nat Commun 2017, 8(1): 1600.

Lee HG, Lee JU, Kim DH, Lim S, Kang I, Choi JM. Pathogenic function of bystander-activated memory-like CD4(+) T cells in autoimmune encephalomyelitis. Nat Commun 2019, 10(1): 709.

Jelcic I, Al Nimer F, Wang J, Lentsch V, Planas R, Jelcic I, et al. Memory B Cells Activate Brain-Homing, Autoreactive CD4(+) T Cells in Multiple Sclerosis. Cell 2018, 175(1): 85-100 e123.

Camperio C, Muscolini M, Volpe E, Di Mitri D, Mechelli R, Buscarinu MC, et al. CD28 ligation in the absence of TCR stimulation up-regulates IL-17A and pro-inflammatory cytokines in relapsing-remitting multiple sclerosis T lymphocytes. Immunol Lett 2014, 158(1-2): 134-142

Point 5: Authors can also clarify if there are any dissimilarity results in the different age group of the patients? 

Response to Point 5: In order to test whether age of patients interferes with the levels of expression of IL-1R1, CCR6, RORgt, and IL-17, we performed statistical correlation between age and level of expression of each protein. We did not find any statistical correlation between age and levels of IL-1R1, CCR6, RORgt, and IL-17. Results are reported in Figure for revision.  We could not perform similar correlation for the other cytokines, where the samples are too few to enable a reliable statistical correlation.

Point 6: Page number 3: Authors could increase the size of the panels of figure 1, the letters looks small.

Response to Point 6: We increased the size of the panels in Figure 1 and also in Figure 2.

Reviewer 2 Report

The authors Capone et al. possess valuable material in terms of samples from MS patients however they fail to raise interesting questions.

Most of their observations are already published by numerous researchers. In addition, the manuscript lacks references to significant and highly-cited articles related to Th17 role in MS. 

Author Response

Reviewer2

Point 1: The authors Capone et al. possess valuable material in terms of samples from MS patients however they fail to raise interesting questions. Most of their observations are already published by numerous researchers.

Response to Point 1: The question we raise was: ”Are Th17 cells generated from MS patients and healthy donors different?”. Given the high importance of Th17 cells in MS, we believe this is an important topic. The originality of our study resides in the systematic analysis of many features of Th17 cells, including cytokines, transcription factors, receptors, that have never been investigated in this model, at protein level. 

Point 2: In addition, the manuscript lacks references to significant and highly-cited articles related to Th17 role in MS. 

Response to Point 2: We thank reviewer for the suggestion and we added significant and highly-cited articles related to Th17 role in MS in the revised version of the manuscript (Line 227-230):

Durelli L, Conti L, Clerico M, Boselli D, Contessa G, Ripellino P, et al. T-helper 17 cells expand in multiple sclerosis and are inhibited by interferon-beta. Ann Neurol 2009, 65(5): 499-509.

Lock C, Hermans G, Pedotti R, Brendolan A, Schadt E, Garren H, et al. Gene-microarray analysis of multiple sclerosis lesions yields new targets validated in autoimmune encephalomyelitis. Nat Med 2002, 8(5): 500-508.

Tzartos JS, Friese MA, Craner MJ, Palace J, Newcombe J, Esiri MM, et al. Interleukin-17 production in central nervous system-infiltrating T cells and glial cells is associated with active disease in multiple sclerosis. Am J Pathol 2008, 172(1): 146-155.

Langrish CL, Chen Y, Blumenschein WM, Mattson J, Basham B, Sedgwick JD, et al. IL-23 drives a pathogenic T cell population that induces autoimmune inflammation. J Exp Med 2005, 201(2): 233-240.
